# Community treatment orders and associations with readmission rates and duration of psychiatric hospital admission: a controlled electronic case register study

Wikus Barkhuizen,[1] Alexis E Cullen,[1] Hitesh Shetty,[2] Megan Pritchard,[2,3] Robert Stewart,[2,3] Philip McGuire,[1,2] Rashmi Patel  [1,2]

¹Department of Psychosis Studies, Institute of Psychiatry, Psychology & Neuroscience, King's College London, London, UK
²NIHR Maudsley Biomedical Research Centre, South London and Maudsley NHS Foundation Trust, London, UK
³Department of Psychological Medicine, Institute of Psychiatry, Psychology & Neuroscience, King's College London, London, UK

**Correspondence to**
Dr Rashmi Patel;
bmj@rpatel.co.uk

## ABSTRACT

**Objectives** Limited evidence is available regarding the effect of community treatment orders (CTOs) on mortality and readmission to psychiatric hospital. We compared clinical outcomes between patients placed on CTOs to a control group of patients discharged to voluntary community mental healthcare.

**Design and setting** An observational study using deidentified electronic health record data from inpatients receiving mental healthcare in South London using the Clinical Record Interactive Search (CRIS) system. Data from patients discharged between November 2008 and May 2014 from compulsory inpatient treatment under the Mental Health Act were analysed.

**Participants** 830 participants discharged on a CTO (mean age 40 years; 63% male) and 3659 control participants discharged without a CTO (mean age 42 years; 53% male).

**Outcome measures** The number of days spent in the community until readmission, the number of days spent in inpatient care in the 2 years prior to and the 2 years following the index admission and mortality.

**Results** The mean duration of a CTO was 3.2 years. Patients receiving care from forensic psychiatry services were five times more likely and patients receiving a long-acting injectable antipsychotic were twice as likely to be placed on a CTO. There was a significant association between CTO receipt and readmission in adjusted models (HR: 1.60, 95% CI 1.42 to 1.80, p<0.001). Compared with controls, patients on a CTO spent 17.3 additional days (95% CI 4.0 to 30.6, p=0.011) in a psychiatric hospital in the 2 years following index admission and had a lower mortality rate (HR: 0.66, 95% CI 0.50 to 0.88, p=0.004).

**Conclusions** Many patients spent longer on CTOs than initially anticipated by policymakers. Those on CTOs are readmitted sooner, spend more time in hospital and have a lower mortality rate. These findings merit consideration in future amendments to the UK Mental Health Act.

## INTRODUCTION

Community treatment orders (CTOs) were introduced in England and Wales under the 2007 Mental Health Act (MHA) to allow for compulsory clinical monitoring of people with serious mental disorders within community care settings and to facilitate recall to a psychiatric hospital following suspected relapse. Patients on a CTO who are recalled to hospital must be reassessed within 72 hours and where longer inpatient treatment is deemed necessary, the CTO may be revoked and the patient may remain admitted as an inpatient under the MHA section through which they were hospitalised prior to being discharged on a CTO. Only patients already hospitalised involuntarily for treatment of a mental disorder are eligible for a CTO. Orders require renewal every 6 months and, if not extended, lapse at which point the patient is effectively discharged from compulsory treatment. Despite initial scepticism from mental health practitioners,[1] their use has exceeded initial government projections[2] with around 5000 currently being issued per year.[3]

Evidence from England and Wales on the effectiveness of CTOs is limited: the only

randomised controlled trial (RCT) found no evidence that CTOs reduce readmission rates[4 5] Two US trials demonstrated varying readmission rates associated with CTOs but these findings were not statistically significant.[6 7] Like the RCT conducted in England and Wales, these trials were unable to include forensic patients, thereby limiting generalisability. A Cochrane review of these RCTs found no support that CTOs reduce the rates of hospital readmissions.[8] Most observational studies conducted in England and Wales[9–11] found support for their use in reducing readmission rates and days spent in hospital. However, these studies were small and underpowered, examined outcomes within the first year only and failed to include a control group. As such, the effects of CTOs on readmission rates and duration of readmission episodes are likely to have been overestimated. Most reviews of international studies conclude that the evidence on the effectiveness of CTOs is mixed and generally does not support improvements in readmission rates or duration of impatient treatment.[1 12–15] A recent meta-analysis of CTO studies concluded that studies without a control condition found evidence for a reduction in readmission rates and bed-days while those with a control condition did not.[16] Evidence from controlled studies on CTOs in England and Wales is currently limited. Due to differences between jurisdictions in healthcare systems, compulsory community treatment programmes and legislation, findings from previous controlled studies in other countries may not be generalisable to England and Wales.

**Strengths and limitations of this study**

► This large observational study of community treatment orders (CTOs) is the first in England and Wales to include an appropriate control group to allow for robust evaluation of outcomes.
► The use of clinical records allowed for evaluation of CTOs in real-world clinical practice, an advantage when it comes to generalisability of the findings.
► The design of this study limited the risk of follow-up bias since all patients within the catchment area could be included in the analyses.
► While we were able to account for several factors associated with the receipt of CTOs, we were not able to account for differences in illness severity between CTO and control patients.

## Aims

We sought to gather data from a large provider of secondary mental healthcare in South London using electronic health record data for all patients who were involuntarily admitted to a psychiatric hospital since the introduction of CTOs in 2008. We aimed to estimate the association of CTOs with hospital readmission by including an appropriate control group of those discharged to voluntary community care while adjusting for the demographic and clinical features associated with CTO receipt. Second, we aimed to compare the number of days spent in inpatient care in the 2 years following the index discharge between patients discharged on CTOs compared with those discharged voluntarily.

## METHODS

### Study population and data collection

Data were obtained from deidentified electronic health records of mental health services provided by the South London and Maudsley National Health Service (NHS) Foundation Trust (SLaM). Serving a catchment of 1.3 million residents in the boroughs of Lambeth, Southwark, Lewisham and Croydon, SLaM provides more than 230 services that include specialist psychosis services, inpatient wards, community and outpatient services. SLaM treats an estimated 5300 inpatients and 45 000 outpatients per year. The Clinical Record Interactive Search (CRIS) system[17 18] collates deidentified patient data from SLaM and contains both structured and free text fields of routine clinical data including case notes and correspondence.[19]

The analysed sample consisted of all SLaM patients who received and completed an involuntary inpatient episode under Section 3 or Section 37 of the MHA between November 2008 and May 2014, allowing for a follow-up period in all cases by using data up to July 2016.

The authors assert that all procedures contributing to this work comply with the ethical standards of the relevant national and institutional committees on human experimentation and with the Helsinki Declaration of 1975, as revised in 2008. All procedures involving human subjects/patients were approved by the CRIS Oversight Committee which is responsible for ensuring all secondary research applications comply with legal requirements and ethical approval obtained from the Oxfordshire Research Ethics Committee C (reference 18/SC/0372).

### Patient and public involvement

No patients were directly involved in the design, planning, conception and conduct of this study.

### Measures

#### Exposure to a community treatment order

The exposure was defined as discharge from an involuntary psychiatric hospital admission under Section 3 or Section 37 of the MHA to community care under a CTO and was compared with a control group of patients discharged to voluntary community care. CTO status and psychiatric hospital admission data were gathered from structured fields within the CRIS dataset. A manual free-text search on 50 random cases was conducted to confirm that no CTO orders were missed using data from these structured fields. A total of 471 (12.9%) control patients and 140 (16.9%) CTO patients were discharged from more than one involuntary psychiatric hospital admission during the study period. A total of 359 patients in the CTO group (43.3%) had been placed on a CTO more than once. For patients who had never been placed on a CTO but had more than one involuntary admission under Section 3 or Section 37 of the MHA, the most recent Section 3 or 37 discharge date was taken as the index date. Where a patient had more than one CTO, the most recent discharge to CTO was taken as the index date.

### Covariates

Demographic and clinical information was gathered to identify factors associated with CTO receipt. Sex, ethnicity, age at the time of the index admission, the mode of involuntary admission under which patients were admitted (either Section 3 issued by an approved mental health practitioner and two doctors or Section 37 issued by criminal courts) and forensic status (determined if individuals were discharged from a forensic ward or to a forensic service) were collected from structured fields. Collected

data included the start and end date of the index admission and the duration of the index admission in days. Psychiatric diagnosis was obtained from both structured fields on CRIS and free text searches to improve the reliability of information at the time of the index episode. Diagnoses were subsequently grouped according to ICD-10 codes.[20]

We extracted the three most recent antipsychotic medications prescribed prior to discharge and method of administration from structured fields. A variable indexing antipsychotic type (oral, long-acting injectable (LAI) or none) was created. Those on more than one antipsychotic were classified in the LAI group if any of the most recently prescribed drugs 2 weeks prior to discharge were administered as LAI (to take into account combined oral and LAI treatment during initiation of LAI). We chose to distinguish between oral and LAI antipsychotics as LAI antipsychotics are associated with different rates of psychiatric hospital readmission compared with oral antipsychotics.

We included the year of study entry as a covariate as a measure of changes in mental healthcare service provision over time which may have been independently associated with risk of psychiatric hospital admission and number of inpatient days.

## Outcomes of community treatment orders

The primary outcome measure was the number of days spent in the community from the discharge date of the index hospital admission until readmission to a psychiatric ward or the end of the follow-up period (31 July 2016). We extracted data on mortality and censored individuals who died during the follow-up period in survival analyses. We additionally measured the number of days spent in inpatient care in the 2 years prior to and the 2 years following the index admission.

## Missing data

There were no missing data for predictor (CTO status) or outcome (readmission/number of inpatient days) variables. The number of patients with missing data for each covariate entered into multivariable analyses was as follows: age: 41; sex: 0; diagnosis: 43; forensic status: 198; antipsychotic route: 107; year of study entry: 0; number of inpatient days in 2 years prior to index admission: 0. Where patients had missing covariate data, they were dropped from multivariable analyses. 92.3% of patients had complete covariate data.

## Analyses

All analyses were conducted in Stata 13[21] and statistical significance was defined at $p < 0.05$. Univariate logistic regression models were first performed to assess the association between CTO status and demographic/clinical factors (sex, age, ethnicity, diagnosis, forensic status, the mode of admission and the route of administration of antipsychotic medication). A fully adjusted model using multivariable logistic regression was subsequently performed with all covariates entered simultaneously to predict CTO status.

To adjust for censoring based on the study follow-up period, mean and median CTO duration were estimated using the 'stci' and 'stsum' commands in STATA censored on 31 July 2016 (the end of the study follow-up window).

Kaplan-Meier curves and log-rank tests were used to assess differences in the observed survival rates (time to next admission) between CTO and voluntarily discharged patients. To assess the association between CTO exposure and time to next admission, unadjusted Cox regression models were performed. Covariates that were significantly associated with CTO exposure in the adjusted logistic regression analyses were then added to the Cox regression in a multivariable model on the basis that factors significantly associated with CTO exposure may be, a priori, most influential on associations with clinical outcomes. Overall significance of the Cox regression model was assessed using the −2 log-likelihood ratio test. A Schoenfeld residuals test and visual inspection of Nelson-Aalen cumulative hazard curves were used to test the proportional-hazards assumption. An additional discrete time analysis was performed where the proportional-hazards assumption was not met.

The CTO and control group were compared on the number of days spent in a psychiatric hospital 2 years after the index date using multiple linear regression with the same covariates as used in the multivariable Cox regression on time to readmission plus an additional covariate measuring the number of days spent in inpatient care 2 years prior to the index admission as this factor was associated with increased number of days spent in a psychiatric hospital subsequently (Pearson coefficient: 0.23, $p < 0.001$).

## RESULTS

The cohort comprised 4489 SLaM patients discharged from involuntary inpatient care between 2008 and 2014. Of these, 830 (18.5%) were placed on a CTO at least once while the remaining 3659 (81.5%) were not. The mean age at discharge of those on CTOs was 39.5 years (SD=13.3) and those discharged without a CTO 42.1 years (SD=16.4). The CTO group had a larger proportion of males compared with those not on a CTO (62.9% vs 52.8%). Table 1 summarises the demographic details of the sample by CTO exposure.

The mean duration of CTO was 3.20 years (95% CI 2.95 to 3.45). The median duration of CTO was 2.65 years (IQR 0.50 to 5.65). Of those whose CTOs ended before the end of the study period (n=434), 19.0% were discharged to standard care, 15.7% had their CTOs revoked and were readmitted to hospital under their previous MHA Section and the CTOs of 12.4% of patients lapsed due to not being actively renewed at the required 6-monthly review.

A total of 489 patients died during the study period. Of these, 432 were in the control group and 57 were in the CTO group. The overall mortality rate was 2.14% per

**Table 1** Sample characteristics and logistic regression of factors associated with CTOs

| | Total group (n=4489) | | Controls (n=3659) | | CTO (n=830) | | Unadjusted models | | | Adjusted models | | |
|---|---|---|---|---|---|---|---|---|---|---|---|---|
| | n | % | n | % | n | % | OR | CI | P value | OR | CI | P value |
| **Sex** | | | | | | | | | | | | |
| Male | 2453 | 54.6 | 1931 | 52.8 | 522 | 62.9 | Ref | – | – | – | – | – |
| Female | 2036 | 45.4 | 1728 | 47.2 | 308 | 37.1 | 0.66 | 0.56 to 0.77 | <0.001 | 0.83 | 0.71 to 0.99 | 0.042 |
| **Age group** | | | | | | | | | | | | |
| <25 | 683 | 15.4 | 577 | 15.8 | 106 | 13.4 | 0.69 | 0.54 to 0.89 | 0.005 | 0.76 | 0.57 to 1.01 | 0.063 |
| 25–34 | 1045 | 23.5 | 826 | 22.6 | 219 | 27.8 | Ref | – | – | – | – | – |
| 35–44 | 1027 | 23.1 | 825 | 22.6 | 202 | 25.6 | 0.92 | 0.75 to 1.14 | 0.466 | 0.91 | 0.72 to 1.15 | 0.415 |
| 45–55 | 880 | 19.8 | 700 | 19.1 | 180 | 22.8 | 0.97 | 0.78 to 1.21 | 0.786 | 0.94 | 0.74 to 1.20 | 0.642 |
| >55 | 813 | 18.3 | 731 | 20.0 | 82 | 10.4 | 0.42 | 0.32 to 0.56 | <0.001 | 0.48 | 0.36 to 0.65 | <0.001 |
| **Ethnicity** | | | | | | | | | | | | |
| White | 1728 | 38.5 | 1449 | 39.6 | 279 | 33.6 | Ref | – | – | – | – | – |
| Black | 2192 | 48.9 | 1727 | 47.2 | 465 | 56.0 | 1.40 | 1.19 to 1.65 | <0.001 | 1.08 | 0.89 to 1.30 | 0.457 |
| Other | 566 | 12.6 | 480 | 13.1 | 86 | 10.4 | 0.93 | 0.72 to 1.21 | 0.591 | 0.79 | 0.59 to 1.07 | 0.134 |
| **Primary diagnosis** | | | | | | | | | | | | |
| Schizophrenia | 2092 | 47.1 | 1609 | 44.0 | 483 | 61.2 | Ref | – | – | – | – | – |
| Other psychotic | 628 | 14.1 | 544 | 14.9 | 84 | 10.7 | 0.51 | 0.40 to 0.66 | <0.001 | 0.60 | 0.46 to 0.79 | <0.001 |
| Bipolar and related | 630 | 14.2 | 554 | 15.2 | 76 | 9.6 | 0.46 | 0.35 to 0.59 | <0.001 | 0.59 | 0.44 to 0.78 | <0.001 |
| Depressive disorders | 428 | 9.6 | 365 | 10.0 | 63 | 8.0 | 0.57 | 0.43 to 0.77 | <0.001 | 0.93 | 0.68 to 1.26 | 0.629 |
| Other | 668 | 15.0 | 585 | 16.0 | 83 | 10.5 | 0.47 | 0.37 to 0.61 | <0.001 | 0.63 | 0.47 to 0.84 | 0.002 |
| **Forensic status** | | | | | | | | | | | | |
| Secondary care | 3983 | 92.8 | 3318 | 95.6 | 665 | 81.1 | Ref | – | – | – | – | – |
| Forensic patient | 308 | 7.2 | 153 | 4.4 | 155 | 18.9 | 5.05 | 3.98 to 6.42 | <0.001 | 4.41 | 3.40 to 5.72 | <0.001 |
| **Route of antipsychotic** | | | | | | | | | | | | |
| Oral | 3162 | 72.2 | 2741 | 74.9 | 421 | 58.2 | Ref | – | – | – | – | – |
| LAI | 1142 | 26.1 | 844 | 23.1 | 298 | 41.2 | 2.30 | 1.94 to 2.72 | <0.001 | 2.18 | 1.83 to 2.61 | <0.001 |
| None | 78 | 1.8 | 74 | 2.0 | 4 | 0.6 | 0.35 | 0.13 to 0.97 | 0.043 | 0.53 | 0.18 to 1.50 | 0.23 |
| **Mode of admission** | | | | | | | | | | | | |
| Section 3 | 4284 | 96.3 | 3546 | 96.9 | 738 | 93.5 | Ref | – | – | – | – | – |
| Section 37 | 164 | 3.7 | 113 | 3.1 | 51 | 6.5 | 2.17 | 1.54 to 3.05 | <0.001 | 1.42 | 0.93 to 2.16 | 0.105 |

Adjusted models accounting for all predictor variables entered simultaneously into multiple logistic models.
CTO, community treatment order; LAI, long-acting injectable.

year. The mortality rate for the control group was 2.27% per year. The mortality rate for the CTO group was 1.53% per year (HR compared with controls 0.66, 95% CI 0.50 to 0.88, p=0.004).

A breakdown of year of study entry is provided in online supplementary table S1 and indicates that rates of CTO usage have varied over time with peak usage occurring in 2013.

### Predictors of receipt of a CTO

In unadjusted logistic regression models (table 1), several significant predictors of CTO exposure were identified. Females had lower odds than males of being placed on a CTO. In those between 25 and 34 years of age more likely to receive a CTO, black patients had higher odds of receiving a CTO compared with white patients. Compared with receiving a diagnosis of schizophrenia, those with other diagnoses had reduced odds of being placed on a CTO. The odds of receiving a CTO were five times higher among those receiving care from forensic services compared with those in secondary (non-forensic) care and 2.3 times higher in those on LAI relative to those on oral antipsychotics. Patients on a Section 37 (typically issued by criminal courts) during their index inpatient episode had twice the odds of

being placed under a CTO compared with patients on a Section 3.

A logistic regression model that adjusted for all predictors (table 1; $\chi^2(15)=320.58$, p<0.001; pseudo-$R^2$=0.08) indicated that all factors reported above, with the exception of patient ethnicity, remained significantly associated with CTOs, although with attenuated ORs.

### Time to readmission

The overall mean time to readmission was 5.55 years (95% CI 5.44 to 5.66). The mean time to readmission for the control group was 5.82 years (95% CI 5.70 to 5.94). The mean time to readmission for the CTO group was 4.02 years (95% CI 3.80 to 5.25). Kaplan-Meier curves (see online supplementary figure S1) comparing the observed survival time to next admission indicated that patients on a CTO were likely to be readmitted sooner than patients who were not on a CTO (univariate HR: 1.76, 95% CI 1.58 to 1.96, p<0.001). A multivariable Cox regression model (table 2) confirmed the association between CTO exposure and reduced time to readmission which remained significant after adjusting for covariates. Figure 1 illustrates the survival rates depending on CTO status. The Schoenfeld residuals test indicated that the proportional hazards assumption was not met ($\chi^2$=14.3, p<0.001) suggesting that the effect of CTO exposure on time next admission was not constant over time. We therefore conducted a discrete time analysis using multivariable logistic regression on discrete periods of 12 months of follow-up (online supplementary table S2) that confirmed the association between CTO receipt and reduced time to next readmission across different periods of follow-up.

### Length of stay

Patients not on a CTO spent a mean of 148.3 days (SD: 164.1) and median of 85 days (IQR 43–183) in a psychiatric hospital in the 2 years after the index date. Patients who were on a CTO spent a mean of 177.7 days (SD: 178.4) and median of 116.5 days (IQR 52–253) in a psychiatric hospital in the 2 years after the index date. After adjusting for covariates in multiple linear regression (table 3), Those on CTOs spent 17.3 additional days (95% CI 4.0 to 30.6) in a psychiatric hospital compared with patients not on a CTO.

Online supplementary figure S2 illustrates that a greater proportion of patients who were not placed on a CTO spent less time in hospital compared with prior to their index admission than patients who were placed on a CTO.

### DISCUSSION
### Main findings

To the best of our knowledge, this is the largest observational study to evaluate the predictors and outcomes of CTOs in England and Wales. Importantly, the inclusion of a control group of patients discharged to voluntary

**Table 2** Multivariable Cox regression analysis of time to next admission following discharge from index admission (n=4144)

| Factor | HR (95% CI) | P value |
|---|---|---|
| Control group (reference) | | |
| CTO group | 1.60 (1.42 to 1.80) | <0.001 |
| Age (reference: <25 years) | | |
| 25–34 years | 1.02 (0.87 to 1.18) | 0.84 |
| 35–44 years | 0.89 (0.76 to 1.04) | 0.13 |
| 45–55 years | 0.80 (0.68 to 0.94) | 0.007 |
| >55 years | 0.65 (0.54 to 0.78) | <0.001 |
| Sex (reference: male) | | |
| Female | 0.95 (0.86 to 1.04) | 0.27 |
| Diagnosis (reference: schizophrenia) | | |
| Other psychotic disorder | 0.90 (0.78 to 1.05) | 0.18 |
| Bipolar disorder | 1.23 (1.07 to 1.41) | 0.004 |
| Depressive disorders | 1.19 (1.01 to 1.41) | 0.041 |
| Other | 0.81 (0.68 to 0.95) | 0.011 |
| Forensic status (reference: not forensic) | | |
| In forensic services | 1.12 (0.94 to 1.34) | 0.22 |
| Antipsychotic route (reference: oral) | | |
| LAI/depot | 1.11 (0.99 to 1.23) | 0.063 |
| None | 0.47 (0.25 to 0.85) | 0.013 |
| Year of study entry (reference: 2008) | | |
| 2009 | 1.20 (0.98 to 1.46) | 0.083 |
| 2010 | 1.29 (1.06 to 1.58) | 0.013 |
| 2011 | 1.47 (1.21 to 1.79) | <0.001 |
| 2012 | 1.81 (1.49 to 2.19) | <0.001 |
| 2013 | 2.02 (1.66 to 2.45) | <0.001 |
| 2014 | 2.45 (1.96 to 3.08) | <0.001 |

Analysis adjusted for all variables reported in this table.
CTO, community treatment order; LAI, long-acting injectable.

care allowed for a robust evaluation of CTOs in real clinical settings. Those placed on CTOs were more likely to be male, in the 25–34 age group, have a diagnosis of schizophrenia-spectrum or depressive disorders, have been treated in forensic mental health services and in receipt of LAI antipsychotics. The greatest proportion of admissions resulting in discharge to a CTO occurred in 2013. CTOs were associated with a higher rate of readmission, even after accounting for associated demographic and clinical features. Patients on a CTO had a greater number of days spent in psychiatric hospital in the 2 years following discharge from index admission than patients not on a CTO.

### Community treatment orders and relapse rates

We found that, at any given time during the follow-up period, those on CTOs had an increased risk of being

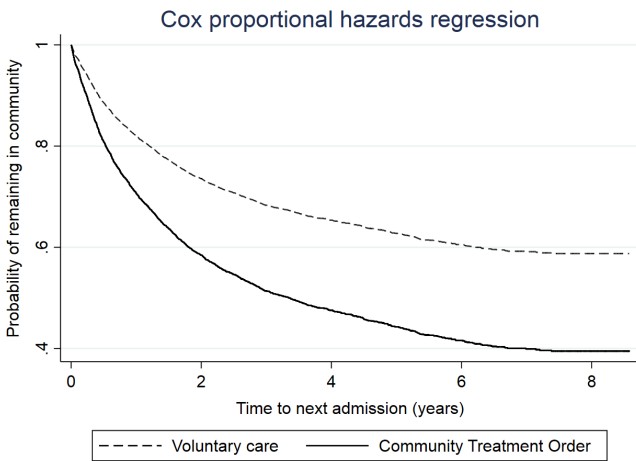

**Figure 1** Graph of Cox survival function comparing time to next admission between CTO and controls (n=4439). CTO, community treatment order.

readmitted to psychiatric inpatient treatment compared with those voluntarily discharged after controlling for factors associated with CTO status. Previous studies in England and Wales[9–11] reported a reduction in hospital days and admission rates after patients were placed on a CTO compared with the 2 years prior to the order. These studies employed uncontrolled before-and-after designs on samples ranging from 20 to 37 patients, all recruited from a single service. Consequently, both statistical power and the generalisability of findings were limited. Also impacting on the generalisability, one study[9] excluded those on short duration CTOs. In the absence of control conditions, effects other than CTO exposure, including the natural trajectory of disease and other treatment effects, are more likely to have accounted for some of the improvement in the rates and duration of readmissions previously reported.

There are several possible explanations for why those on CTOs had higher rates of psychiatric hospital admission compared with controls in our study. One plausible explanation is that patients with more severe symptoms or a history of relapse were more likely to receive CTOs. Hospitalisation rates may also be higher due to the relative ease of readmission of deteriorating patients under the CTO pathway, which does not require a MHA assessment to be conducted. An alternative possibility is that CTOs were not effective at reducing relapse rates. While we controlled for factors associated with receipt of CTOs (forensic status, diagnosis and the route of antipsychotic administration), we were not able to account for possible differences in illness severity between CTO patients and controls, nor did we compare relapse rates prior to and after being placed on CTOs.

Our study also examined CTO outcomes in terms of differences in the number of days spent in psychiatric hospitals. After adjusting for covariates including the amount of time spent in a psychiatric hospital prior to index admission, those on CTOs spent 17 additional

**Table 3** Multiple linear regression investigating number of inpatient days in a psychiatric hospital during the 2 years following discharge from index admission (n=4190)

| Factor | β coefficient (95% CI) | P value |
|---|---|---|
| Control group (reference) | | |
| CTO group | 17.3 days (4.0 to 30.6) | 0.011 |
| Age (reference: <25 years) | | |
| 25–34 years | −6.5 days (−22.4 to 9.5) | 0.43 |
| 35–44 years | −23.7 days (−39.7 to −7.7) | 0.004 |
| 45–55 years | −12.6 days (−29.2 to 4.0) | 0.14 |
| >55 years | 13.9 days (−2.9 to 30.8) | 0.11 |
| Sex (reference: male) | | |
| Female | −17.4 days (−27.2 to −7.7) | <0.001 |
| Diagnosis (reference: schizophrenia) | | |
| Other psychotic disorder | −21.1 days (−35.5 to −6.7) | 0.004 |
| Bipolar disorder | −40.2 days (−54.6 to −25.8) | <0.001 |
| Depressive disorders | −33.6 days (−50.5 to −16.7) | <0.001 |
| Other | 28.7 days (13.7 to 43.8) | <0.001 |
| Forensic status (reference: not forensic) | | |
| In forensic services | 33.4 days (14.1 to 52.7) | 0.001 |
| Antipsychotic route (reference: oral) | | |
| LAI/depot | −15.8 days (−26.8 to −4.8) | 0.005 |
| None | 29.0 days (−9.0 to 67.0) | 0.14 |
| Year of study entry (reference: 2008) | | |
| 2009 | −13.0 days (−31.4 to 5.4) | 0.17 |
| 2010 | −9.2 days (−27.6 to 9.2) | 0.33 |
| 2011 | −12.6 days (−30.7 to 5.4) | 0.17 |
| 2012 | −17.6 days (−35.1 to −0.1) | 0.049 |
| 2013 | −3.8 days (−21.2 to 13.6) | 0.67 |
| 2014 | −3.5 days (−24.9 to 18.0) | 0.75 |
| Number of inpatient days in 2 years prior to start of index admission | 0.31 days (0.27 to 0.35) | <0.001 |

Analysis adjusted for all variables reported in this table.
CTO, community treatment order; LAI, long-acting injectable.

days in a psychiatric hospital during the follow-up period compared with patients who were not placed on a CTO. Our findings are in line with previous studies that have examined the number of days spent in hospital pre-CTO and post-CTO exposure;[9–11] however, these studies failed to include a control group.

### The duration of community treatment orders
We report that, once a CTO is issued, many patients are kept under this treatment option for extensive periods. The mean duration of a CTO in our study was 3 years, far exceeding initial government projections of 9 months.[22]

This is informative because while official figures on the number of CTOs issued per year are readily available,[3] national statistics on their duration are hard to come by. In the current sample, CTOs also lasted longer than previously found in samples from England and Wales. For instance, we found that CTOs lasted three times as long compared with Dye et al[11] who reported an average duration of 52.6 weeks (SD=31.7). A likely reason for this may be the longer follow-up period in this study. At the end of Dye et al's 2-year follow-up period, 35% of their sample were still on CTOs. That a large proportion of the eligible patient population are exposed to compulsory community treatment, often for long periods of time, underscores the importance of understanding the characteristics of and outcomes for those on CTOs.

### Demographic and clinical features of patients who receive community treatment orders

The age and sex distributions within the CTO subsample were similar to previous reports.[10 11 23] Unlike previous studies, we were able to compare the demographics of those on CTOs to those discharged without a CTO: taking into account other factors, those already hospitalised under the MHA were 4.4 times more likely to receive a CTO if they were in forensic services, twice as likely to be on LAI than oral antipsychotics, more likely male and with a diagnosis of a schizophrenia-spectrum disorder compared with bipolar, other psychotic or miscellaneous disorders (but not depression).

Almost 20% of those placed on CTOs were either hospitalised within forensic wards or discharged to forensic services. Consequently, those with forensic service involvement were more likely than non-forensic patients to be subject to compulsory treatment. Considering that the most commonly cited reasons why patients are placed on CTOs in England and Wales relates to clinical considerations such as improving adherence to treatment,[24] as opposed to monitoring risk, this result is somewhat surprising. It is possible that CTOs may have been used primarily for protective or monitoring reasons among forensic patients more commonly than previously suggested. In the absence of further investigation into the motivations for issuing CTOs and their conditions within this sample, it is not clear what underlies the reason that forensic status was found to be predictive of being placed on a CTO.

The proportion of patients of black ethnicity being placed on CTOs was similarly high to that reported by Patel et al[23] in their cohort within the same geographic area of London. Patel et al speculated that their finding may have been due to the high number of black patients sectioned under the MHA. The current study was able to explore this phenomenon in further detail as it included all those detained under the MHA and found that black patients who were already in psychiatric hospitals were significantly more likely to be placed on a CTO than other patients. However, the association between ethnicity and CTO exposure did not remain significant after considering other predictors of being started on a CTO, including forensic, clinical and demographic characteristics, suggesting that these factors explained the positive association we observed between black ethnicity and CTO receipt.

Like others, this study found that a large proportion of those on CTOs were prescribed LAI formulation antipsychotics (41%).[9 10 23 25] Moreover, this study demonstrated that those on a CTO had twice the odds of being prescribed LAIs than those not on a CTO, a finding that remained significant after controlling for the effect of diagnosis. Several reasons for a preference by psychiatrists for prescribing LAI formulation antipsychotics to those on CTOs have been proposed: Patel et al[23] suggested that those less likely to adhere are placed on LAI and therefore at greater risk of relapse. This would make them likely candidates for a CTO. Adherence to LAIs is also easier to monitor and therefore preferable for reasons of enforcement.

The significantly lower mortality rate among those who received CTOs during our study period is in line with similar reports that compared mortality rates between CTO and control patients in Australia.[26 27] Kisely et al[26] suggested that the differences in mortality rates in their study may be due to closer follow-up in the community among patients on CTOs. This may enable better detection and treatment of emerging physical health disorders and highlights the importance of robust community follow-up for patients following discharge from psychiatric hospital regardless of CTO status.

### Strengths and limitations

The advantage of using health records to investigate the use and effects of CTOs is that they provide data which represents real-world clinical practice which may be more generalisable than findings from RCTs whose participants may not be representative of the wider population receiving mental healthcare. Additionally, follow-up bias is limited as most subjects are included at all time points of data gathering unless they leave the area or die.[28 29] The SLaM BRC CRIS system has specific advantages over many other electronic case registers: with access to over 400 000 patient records and combining structured fields and free-text searches from fully digitised case registers, it has both the quality of smaller databases and the quantity of data only large registers provide.[17] On the other hand, the use of routine clinical data limits the possible number of covariates that can be reliably controlled for and does not rule out confounding by indication. For instance, this study did not control for illness severity, urbanicity, migration status, educational attainment or socioeconomic characteristics. Electronic patient records are also susceptible to erroneous or partial recording of treatment events. During the study period, patients may have moved out of the catchment area or into the catchment area from elsewhere. As the data source only included records from SLaM, CTO and hospital admission episodes in

other mental healthcare providers would not have been included.

Using the most recent CTO episode that patients were subject to (in cases where the same patient was subject to more than one CTO during the window period) enabled the study to report on current practices; the way in which CTOs have been used in other jurisdictions has indicated possible change over time as practitioners become more familiar with the legislation.[30]

The rates of readmission reported here should be interpreted with some cautions as our survival analysis did not meet the assumption of proportional hazards. However, discrete time analysis indicated that the association of CTOs with increased rates of readmission persisted across all periods of follow-up.

## Implications

There are several implications for both current and future research into the effects of CTOs. We found that patients spend several years on a CTO, far longer than the original projected duration of 9 months. At the same time, our study demonstrated that patients on CTOs have greater rates of readmission and spend longer in psychiatric hospital than patients who are not on CTOs, going against previously published studies which did not include a control group.

We found that certain demographic and clinical features, such as ethnicity and forensic background, are strongly associated with CTO exposure. Several observations on the use of CTOs warrant further investigation, including the influence of LAIs on outcomes, and the effects of extensive periods of community compulsion on patient engagement.

We found a higher incidence of CTOs among black patients adding to previous findings which indicate that black and minority ethnic groups are over-represented in compulsory mental healthcare admission and treatment.

In conclusion, our study found that patients may spend several years on a CTO and that CTOs are not associated with a reduction in psychiatric hospital admission or with less time spent in a psychiatric hospital. The implementation of CTOs in future amendments of the UK MHA should be reviewed in light of these findings.[31]

**Contributors** The study was conceived by AEC and RP. Data extraction was performed by WB with support from HS and MP. Statistical analyses and reporting of findings were carried out by WB, supervised by AEC and RP. WB, AEC, HS, MP, RS, PM and RP contributed to study design, manuscript preparation and approved the final version.

**Funding** HS, MP, RS and PM receive funding from the National Institute for Health Research (NIHR) Biomedical Research Centre at South London and Maudsley NHS Foundation Trust and King's College London, which also supports the development and maintenance of the BRC Case Register. RP has received support from a Medical Research Council (MRC) Health Data Research UK Fellowship (MR/S003118/1) and a Starter Grant for Clinical Lecturers (SGL015/1020) supported by the Academy of Medical Sciences, The Wellcome Trust, MRC, British Heart Foundation, Arthritis Research UK, the Royal College of Physicians and Diabetes UK. AEC has received support from a Sir Henry Wellcome Postdoctoral Fellowship (107395/Z/15/Z).

**Disclaimer** The views expressed are those of the authors and not necessarily those of the NHS, the NIHR or the Department of Health. The funders had no role in the design and conduct of the study; collection, management, analysis and interpretation of the data; preparation, review or approval of the manuscript and decision to submit the manuscript for publication.

**Competing interests** None declared.

**Patient consent for publication** Not required.

**Ethics approval** The CRIS data resource received ethical approval as an anonymised dataset for secondary analyses from Oxfordshire REC C (Ref: 08/H0606/71+5).

**Provenance and peer review** Not commissioned; externally peer reviewed.

**Data availability statement** Data are available on reasonable request. The data accessed by CRIS remain within an NHS firewall and governance is provided by a patient-led oversight committee. Subject to these conditions, data access is encouraged and those interested should contact RS (robert.stewart@kcl.ac.uk), CRIS academic lead.

**ORCID iD**
Rashmi Patel http://orcid.org/0000-0002-9259-8788

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
