## [Reviewer comments · BMJ Open]

ARTICLE DETAILS

TITLE (PROVISIONAL)	Community Treatment Orders and associations with readmission rates and duration of psychiatric hospital admission: A controlled electronic case register study
AUTHORS	Barkhuizen, Wikus; Cullen, Alexis E.; Shetty, Hitesh; Pritchard, Megan; Stewart, Robert; McGuire, Philip; Patel, Rashmi

VERSION 1 – REVIEW

REVIEWER	Tom Burns University of Oxford, UK
REVIEW RETURNED	28-Oct-2019

GENERAL COMMENTS	This is a wonderful database and its comprehensiveness allows for much more confidence in the interpretation of the findings than is so often the case. The power of the database and the sophistication of the statistical analysis appear to drive the paper at the expense of the clinical questions being asked and make it slightly frustrating to read as a clinician. In addition the presentation of the results and the discussion being rather 'mechanical' may obscure important findings. The most obvious and important findings of this study would appear to be that CTOs tend to be long term, they are associated with increased risk of readmission and longer stays than not being on a CTO (the opposite of their anticipated effect) that they are very frequently used by Forensic services. In addition the strikingly higher rates in black patients are fully accounted for by other baseline characteristics. There are a number of points that would benefit from attention. The literature review is rather patchy and surprising. For instance there are three RCTs but only one is mentioned. Where Swartz and Stanton are cited it is in an opinion piece that flatly contradicts their own RCT findings. Similarly, while this study may be the first large controlled cohort study in the UK Segal has conducted an even larger one in Australia. The benefits of citing that may lie in many of the obvious criticisms it attracted may also apply here. The OCTET follow up study also pointed at the duration of CTOs which should come as no surprise (despite the DH impact statement) if practice in Australia and New Zealand had been surveyed and cited. OCTET exclusion of Forensic patients was for administrative reasons. The concerns about its generalizability expressed here could perhaps be addressed by checking whether or not there were any significant differences in the outcomes between Forensic
---

	and non-Forensic patients. That would be interesting in its own right quite apart from the comment on OCTET. OCTET also excluded patients above 65 years (for similar reasons) and there are clinical questions being raised about CTOs in this group. It is unclear from the data presentation if those above 65 were included. Again, if they are then there is the potential for some interesting exploration. P9 Quotes a recent meta-analysis confirming that non controlled studies find a benefit but controlled ones do not and then continues "However, due to differences between jurisdictions in health care systems and compulsory community treatment programmes and legislation, results may not be generalisable to England and Wales". I can see no logic to this "however". Indeed it strikes me as remarkable that non controlled studies and controlled studies are reported as if they were of equal value. p17 The reporting of status at ending of the CTO is a bit confusing here p18 is the lower mortality rate in CTO patients not worth some comment? p31 the reporting that ethnic variation is explained by other baseline factors ought to be emphasised a bit given how much controversy this has, understandably, generated. p31. It is incorrect to state ts 'In conclusion, our study demonstrates that the potential benefits and harms of CTOs remain unclear'. This is essentially a descriptive study and cannot confidently draw conclusions about effects. If it could the conclusion would be that CTOs are basically bad for outcomes (ie more rapid relapse and longer in hospital) but it is clear that that in the absence of being able to control for severity of disorder at intake no such conclusion can be drawn.
--	---

REVIEWER	Miharu Nakanishi Tokyo Metropolitan Institute of Medical Science, Japan
REVIEW RETURNED	03-Nov-2019

GENERAL COMMENTS	The present study examined outcomes of Community Treatment Orders in comparison with a control condition. The implications from the results may add to improvement for mental health care systems. Two major questions would be addressed:  1. Analyses: There may have been argument about variable selection of multivariate regression based on significance in univariate regression. Although the sample size was large (830 v.s. 3659), this approach would be subject to inconsistency across different populations. Recent statistical recommendations may suggest use of all covariates as planned in multivariate regression. Please consider (1) adding rationales for variable selection to the manuscript, or (2) entering all covariates in multivariate regression. 2. As the authors mentioned, the change between pre- and post-intervention is not equal to the difference between intervention and control group. A simplified interpretation of results would be
--

	inserted into Discussion. What is implications for the population who have received CTOs, should they be treated in voluntary community care or another form of coercive care instead of CTOs?
--	--

REVIEWER	Steve Kisely UQ, Australia
REVIEW RETURNED	05-Nov-2019

GENERAL COMMENTS	This interesting paper reports on an observational study of de-identified electronic health record data from inpatients receiving mental healthcare in South London using the Clinical Record Interactive Search (CRIS) system. Patients discharged from compulsory inpatient treatment between November 2008 and May 2014 were considered to compare outcomes for those on compulsory & voluntary community treatment over a two-year follow-up period. They found that many patients spent longer on CTOs than initially anticipated by policy makers & that those on CTOs were readmitted sooner and spent more time in hospital on multivariate analyses. The paper would benefit from clarification/ acknowledgement of the following issues:  1) Did they follow appropriate guidelines for conduct of the study such as the STrengthening the Reporting of OBServational studies in Epidemiology (STROBE)? 2) The authors were able to adjust for a range of clinical and demographic factors. However, they failed to acknowledge that they were unable to adjust for important variables such as level of education, employment status, country of birth and social economic status. As regards the latter, is it possible to infer this from residential address and census enumeration district data? At the very least, these limitations should be acknowledged in the discussion as they may have been important potential confounders. 3) One important finding is not discussed at all, including a mention in the abstract. This is the fact that mortality was significantly less for the CTO group compared to the controls. This needs greater consideration and mirrors findings from the following two Australian studies.  - Segal SP, Burgess PM. Effect of conditional release from hospitalization on mortality risk. Psychiatr Serv. 2006 Nov;57(11):1607-13. - Kisely S, Preston N, Xiao J, Lawrence D, Louise S, Crowe E. Reducing all-cause mortality among patients with psychiatric disorders: a population-based study. CMAJ. 2013 Jan 8;185(1):E50-6. doi: 10.1503/cmaj.121077. Epub 2012 Nov 12. There are two possible explanations for this finding. One is that community treatment orders allow for adequate supervision of care including physical care and so this is an example of the success of community treatment orders. Another, is that this advantage disappears if findings are adjusted for the level of outpatient or community contacts, and that there may be alternative ways of achieving this rather than compulsory community treatment. This discussion merits greater consideration in the present paper. 4) On a small note, the authors do not need to include both the Kaplan-Meier & Cox regression curves. One or other is perfectly adequate.
--

	5) Some of the references are quite old and don't include relevant recent systematic reviews of the area.
--	---

VERSION 1 – AUTHOR RESPONSE

Reviewer: 1

Reviewer Name: Tom Burns

Institution and Country: University of Oxford, UK Please state any competing interests or state 'None declared': none

This is a wonderful database and its comprehensiveness allows for much more confidence in the interpretation of the findings than is so often the case.

/*Thank you for your supportive comments on our manuscript. We have responded to your suggestions for amendment below.*/

The power of the database and the sophistication of the statistical analysis appear to drive the paper at the expense of the clinical questions being asked and make it slightly frustrating to read as a clinician. In addition the presentation of the results and the discussion being rather 'mechanical' may obscure important findings. The most obvious and important findings of this study would appear to be that CTOs tend to be long term, they are associated with increased risk of readmission and longer stays than not being on a CTO (the opposite of their anticipated effect) that they are very frequently used by Forensic services. In addition the strikingly higher rates in black patients are fully accounted for by other baseline characteristics.

/*We appreciate the importance of presenting our findings to a clinical readership and have updated the wording in our manuscript to highlight the main key findings which include the long duration of CTOs, association with increased risk of readmission and longer inpatient stay and use in forensic services. We hope this will make the manuscript more accessible to clinicians and healthcare policy makers, particularly in light of ongoing reforms to the UK Mental Health Act and to mental health legislation in other countries.*/

There are a number of points that would benefit from attention.

The literature review is rather patchy and surprising. For instance there are three RCTs but only one is mentioned. Where Swartz and Stanton are cited it is in an opinion piece that flatly contradicts their own RCT findings. Similarly, while this study may be the first large controlled cohort study in the UK Segal has conducted an even larger one in Australia. The benefits of citing that may lie in many of the obvious criticisms it attracted may also apply here. The OCTET follow up study also pointed at the duration of CTOs which should come as no surprise (despite the DH impact statement) if practice in Australia and New Zealand had been surveyed and cited.

/*We appreciate the limitations of our literature review which we initially restricted to relevant studies conducted in England and Wales for reasons of brevity within an original research manuscript and for relevance to our study. We have now added additional references to the introduction section including the two US RCTs and the OCTET follow-up study to provide a more comprehensive review of the literature.

We cite a recent meta-analysis by Barnett et al. (2018) which included the Segal et al. study. In addition, we have now included citations of earlier systematic reviews and meta-analyses to the introduction.

We agree that the finding that CTOs in our study exceeded initial government projections are unsurprising and cite previous reports that found similar findings. Our finding that CTOs in the catchment area we studied lasted ~3 years is informative given the long duration of our follow-up period compared to previous studies.*/

OCTET exclusion of Forensic patients was for administrative reasons. The concerns about its generalizability expressed here could perhaps be addressed by checking whether or not there were any significant differences in the outcomes between Forensic and non-Forensic patients. That would be interesting in its own right quite apart from the comment on OCTET. OCTET also excluded patients above 65 years (for similar reasons) and there are clinical questions being raised about CTOs in this group. It is unclear from the data presentation if those above 65 were included. Again, if they are then there is the potential for some interesting exploration.

/*Our findings indicate that receiving care from a forensic mental health service did not significantly affect the time to next admission when comparing CTO and voluntary patients (adjusted hazard ratio = 1.12, 95% CI 0.94 – 1.34; Table 2). CTO patients spent on average 33.4 days longer in psychiatric hospital during our follow-up period compared to controls, a difference that was significant after adjustment for other confounders (p = .001; Table 3). We agree that comparing forensic and non-forensic services would be an interesting research question for further exploration but our study was not designed to address this question and would require a different approach to data extraction and analysis which is beyond the scope of the data extracted for our study.

We did not exclude anyone above 65 years of age from the analyses. We agree that it would be useful to examine whether there are any differences in the use of and outcomes related to CTOs in older adults. Fewer than 5% of CTO patients in our study were aged 65 years or older which limits our statistical power to explore outcomes for this subsample. However, we did examine differences in patients over the age of 55 years and found that these patients were significantly less likely to receive a CTO even after accounting for other predictors of CTO status.*/

P9 Quotes a recent meta-analysis confirming that non controlled studies find a benefit but controlled ones do not and then continues "However, due to differences between jurisdictions in health care systems and compulsory community treatment programmes and legislation, results may not be generalisable to England and Wales". I can see no logic to this "however". Indeed it strikes me as remarkable that non controlled studies and controlled studies are reported as if they were of equal value.

/*We agree that this wording is unclear given that our data are drawn from a large provider of NHS mental healthcare in England. The point we were trying to make was that there are inconsistencies between findings from controlled and uncontrolled studies, the former being more reliable as discussed in the preceding sentences of the same section. We highlight that no similarly robust controlled studies have been performed in England and Wales. We have now rephrased this sentence to read: "Evidence from controlled studies on CTOs in England and Wales is currently lacking. Due to differences between jurisdictions in health care systems, compulsory community treatment programmes and legislation, findings from previous controlled studies in other countries may not be generalisable to England and Wales."*/

p17 The reporting of status at ending of the CTO is a bit confusing here

/*We have rephrased this section and added some additional clarification to our reporting of the different CTO discharge reasons as follows: "Of those whose CTOs ended before the end of the study period (n = 434), 19.0% were discharged to standard care, 15.7% had their CTOs revoked and were readmitted to hospital under their previous MHA Section, and the CTOs of 12.4% of patients lapsed due to not being actively renewed at the required 6-monthly review."*/

p18 is the lower mortality rate in CTO patients not worth some comment?

/*We thank the reviewer for highlighting that we omitted further comment on this finding and have now discussed the lower mortality rates in CTO patients vs controls in the discussion on p19 as follows: "The significantly lower mortality rate among those who received CTOs during our study period is in line with similar reports that compared mortality rates between CTO and control patients in Australia. Kisely et al suggested that the differences in mortality rates in their study were largely due to enhanced service engagement among CTO patients."*/

p31 the reporting that ethnic variation is explained by other baseline factors ought to be emphasised a bit given how much controversy this has, understandably, generated.

/*In line with previous reports, it was clear from our study that those of black Caribbean and African origin were disproportionately subject to compulsory treatment in the community. We dedicate a paragraph in the discussion on p18 to point out that our findings indicate that factors other than ethnicity influenced the decision by clinicians to issue a CTO in the current sample.*/

p31. It is incorrect to state ts 'In conclusion, our study demonstrates that the potential benefits and harms of CTOs remain unclear'. This is essentially a descriptive study and cannot confidently draw conclusions about effects. If it could the conclusion would be that CTOs are basically bad for outcomes (ie more rapid relapse and longer in hospital) but it is clear that that in the absence of being able to control for severity of disorder at intake no such conclusion can be drawn.

/*We agree that this sentence may have been misleading given the nature of our study design and have now removed this statement from the manuscript.*/

Reviewer: 2

Reviewer Name: Miharu Nakanishi

Institution and Country: Tokyo Metropolitan Institute of Medical Science, Japan Please state any competing interests or state 'None declared': None declared

The present study examined outcomes of Community Treatment Orders in comparison with a control condition. The implications from the results may add to improvement for mental health care systems. Two major questions would be addressed:

1. Analyses: There may have been argument about variable selection of multivariate regression based on significance in univariate regression. Although the sample size was large (830 v.s. 3659), this approach would be subject to inconstancy across different populations. Recent statistical recommendations may suggest use of all covariates as planned in multivariate regression. Please consider (1) adding rationales for variable selection to the manuscript, or (2) entering all covariates in multivariate regression.

/*We agree with the reviewer that there is debate around whether to include all available predictors in multivariable analyses. To clarify, we did include all available predictors in multivariable regression models to predict whether patients were discharged on a CTO or not (as per the "Adjusted models" column in Table 1). Based on the findings presented in Table 1, we chose to include only factors that were associated with CTO status in subsequent analyses on clinical outcomes on the basis that these factors may be, *a priori*, most influential on associations with clinical outcomes. We feel this approach provides a more meaningful interpretation of the dataset based on the underlying clinical and demographic factors which we found were significantly associated with CTO status and have updated the methods section to justify our approach.*/

2. As the authors mentioned, the change between pre- and post-intervention is not equal to the difference between intervention and control group. A simplified interpretation of results would be inserted into Discussion. What is implications for the population who have received CTOs, should they be treated in voluntary community care or another form of coercive care instead of CTOs?

/*We agree that a before-and-after study is not directly comparable with a controlled observational study. We have updated the conclusion section of the discussion in response to Professor Burns to provide a clearer interpretation of the main findings. However, our study was observational in nature and we do not feel that our results can provide a straightforward recommendation about whether CTOs should be used or not as this decision depends on a range of clinical and practical factors which vary between individual patients and require careful consideration prior to considering compulsory treatment.*/

Reviewer: 3

Reviewer Name: Steve Kisely

Institution and Country: UQ, Australia

Please state any competing interests or state 'None declared': None declared

This interesting paper reports on an observational study of de-identified electronic health record data from inpatients receiving mental healthcare in South London using the Clinical Record Interactive Search (CRIS) system. Patients discharged from compulsory inpatient treatment between November 2008 and May 2014 were considered to compare outcomes for those on compulsory & voluntary community treatment over a two-year follow-up period. They found that many patients spent longer on CTOs than initially anticipated by policy makers & that those on CTOs were readmitted sooner and spent more time in hospital on multivariate analyses.

The paper would benefit from clarification/ acknowledgement of the following issues:

1) Did they follow appropriate guidelines for conduct of the study such as the STrengthening the Reporting of OBServational studies in Epidemiology (STROBE)?

Yes, and we have enclosed a completed STROBE checklist with our resubmission as requested by the editors.

2) The authors were able to adjust for a range of clinical and demographic factors. However, they failed to acknowledge that they were unable to adjust for important variables such as level of education, employment status, country of birth and social economic status. As regards the latter, is it possible to infer this from residential address and census enumeration district data? At the very least, these limitations should be acknowledged in the discussion as they may have been important potential confounders.

/*We have now included a more detailed discussion on the likely confounders that our study was unable to account for in the limitations section as follows: "...the use of routine clinical data limits the possible number of covariates that can be reliably controlled for... For instance, this study did not control for illness severity, urbanicity, migration status, educational attainment or socioeconomic characteristics."*/

3) One important finding is not discussed at all, including a mention in the abstract. This is the fact that mortality was significantly less for the CTO group compared to the controls. This needs greater consideration and mirrors findings from the following two Australian studies.

- Segal SP, Burgess PM. Effect of conditional release from hospitalization on mortality risk. *Psychiatr Serv.* 2006 Nov;57(11):1607-13.
- Kisely S, Preston N, Xiao J, Lawrence D, Louise S, Crowe E. Reducing all-cause mortality among patients with psychiatric disorders: a population-based study. *CMAJ.* 2013 Jan 8;185(1):E50-6. doi: 10.1503/cmaj.121077. Epub 2012 Nov 12.

There are two possible explanations for this finding. One is that community treatment orders allow for adequate supervision of care including physical care and so this is an example of the success of community treatment orders. Another, is that this advantage disappears if findings are adjusted for the level of outpatient or community contacts, and that there may be alternative ways of achieving this rather than compulsory community treatment. This discussion merits greater consideration in the present paper.

/*We agree that this is an important finding and have updated the manuscript to discuss this further. We have now updated the abstract to include the mortality results and added a section to the discussion and cited the relevant literature as follows: "The significantly lower mortality rate among those who received CTOs during our study period is in line with similar reports that compared mortality rates between CTO and control patients in Australia. Kisely et al suggested that the differences in mortality rates in their study may be due to closer follow-up in the community among patients on CTOs. This may enable better detection and treatment of emerging physical health disorders and highlights the importance of robust community follow-up for patients following discharge from psychiatric hospital regardless of CTO status."*/

4) On a small note, the authors do not need to include both the Kaplan-Meier & Cox regression curves. One or other is perfectly adequate.

/*We agree with the reviewer that one or the other graph is adequate and include only the Cox regression curve that represent survival analyses after adjustment for confounders in our manuscript.

We include the Kaplan-Meier curve (unadjusted) in the supplement for those who may want to see the unadjusted survival curves.*/

5) Some of the references are quite old and don't include relevant recent systematic reviews of the area.

/*We have added citations to additional systematic reviews and meta-analyses to the introduction. In addition to studies performed in England and Wales, we originally cited one meta-analysis on the outcomes of CTOs as this was the most recent review of relevant studies to the best of our knowledge: Barnett P, Matthews H, Lloyd-Evans B, et al. Compulsory community treatment to reduce readmission to hospital and increase engagement with community care in people with mental illness: a systematic review and meta-analysis. Lancet Psychiatry 2018;5(12):1013-22*/

VERSION 2 – REVIEW

REVIEWER	tom burns university of oxford uk
REVIEW RETURNED	27-Nov-2019

GENERAL COMMENTS	This is now much improved by having a clearer description of the aims and by having conclusions that can be confidently derived from the data. I am happy for the article to be published as is but would mention three minor points that could be addressed: p9 line 48 replace 'do' with 'does' p9 My reading of the two US RCTs is that they do not find contrary conclusions. Neither found a significant difference although the rates did differ. p10 I fail to understand the sentence 'evidence for controlled studies in E&W is lacking'. Do the authors mean 'limited' as in page 9? One of the only 3 RCTs was conducted in England (see page 9 lines7-10).
---

REVIEWER	Miharu Nakanishi Tokyo Metropolitan Institute of Medical Science, Japan
REVIEW RETURNED	04-Dec-2019

GENERAL COMMENTS	Thank you for your revision in response to previous comments. I confirmed that my questions have been addressed in the revised version.
---

REVIEWER	Steve Kisely UQ
REVIEW RETURNED	26-Nov-2019

GENERAL COMMENTS	I believe the authors have addressed all the reviewers' comments
--